

# Two new submodels for the Modular Earth Submodel System (MESSy): New Aerosol Nucleation (NAN) and small ions (IONS) version 1.0

Sebastian Ehrhart[1], Eimear M. Dunne[2,3], Hanna E. Manninen[4], Tuomo Nieminen[5], Jos Lelieveld[1], and Andrea Pozzer[1]

[1]Max Planck Institute for Chemistry, Hahn-Meitner-Weg 1, 55128 Mainz, Germany
[2]Department of Geography, University of Cambridge, Cambridge, United Kingdom
[3]Finnish Meteorological Institute, Atmospheric Research Centre of Eastern Finland, PO Box 1627, 70211 Kuopio, Finland
[4]Experimental Physics Department, CERN, 1211 Geneva, Switzerland
[5]Department of Applied Physics, University of Eastern Finland, PO Box 1627, 70211 Kuopio, Finland

*Correspondence to:* Sebastian Ehrhart (s.ehrhart@mpic.de)

**Abstract.** Two new submodels for the Modular Earth Submodel System (MESSy) were developed. The New Aerosol Nucleation submodel (NAN) includes new parameterisations of aerosol particle formation rates published in recent years. These parameterisations include ion-induced nucleation and nucleation of pure organic species. NAN calculates the rate of new particle formation based on the aforementioned parameterisations for aerosol submodels in the ECHAM/MESSy Atmospheric chemistry - Climate (EMAC) model. The Ion pair production rate, needed to calculate the ion-induced or -mediated nucleation, is described using the new submodel IONS, which provides ion pair production rates for other submodels within the MESSy framework. Both new submodels were tested in EMAC simulations. These simulations showed good agreement with ground based observations.

## 1 Introduction

The influence of aerosol particles on various aspects of climate and human health (Knibbs et al., 2011; Lelieveld et al., 2015) is well established. Aerosol particles influence climate through aerosol-cloud and the aerosol-radiation interactions (Lohmann et al., 2010). A detailed understanding of the sources of aerosol particles is necessary to study their climate and health effects. New Particle Formation (NPF), i.e. nucleation and growth of new aerosol particles from vapours, is an important source of secondary aerosol particles in the troposphere and planetary boundary layer and observed events of NPF are well documented (Weber et al., 1999; Kulmala et al., 2004). Manninen et al. (2010) give examples of NPF at various European measurement sites, Pierce et al. (2014) in Canada, Bae et al. (2010) in the USA, Suni et al. (2008) in Australia and Sipilä et al. (2016) observed NPF in a coastal region of Ireland. According to Merikanto et al. (2009) and Yu and Luo (2009) a significant proportion, about 50% globally, of Cloud Condensation Nuclei (CCN) originate from NPF.

Many global model studies of atmospheric aerosols rely on the Binary Homogeneous Nucleation (BHN) parameterisation of Vehkamäki et al. (2002), which describes aerosol particle nucleation using a polynomial fit to a microphysical model of





nucleation as function of $H_2SO_4$ concentration, temperature and relative humidity. Yu (2010) and Kazil et al. (2010) published look up tables for a nucleation parameterisation that includes the effect of airborne ions, Ion Mediated Nucleation (IMN) and Ion Induced Nucleation (IIN) respectively. Ball et al. (1999) showed that $NH_3$ can enhance nucleation rates in a mixture with $H_2SO_4$ and water vapour. Merikanto et al. (2007) derived a first parameterisation of the $H_2SO_4$-$NH_3$-$H_2O$ system based on

theoretical calculations. However, observed boundary layer nucleation rates can not be explained by $H_2SO_4$-$NH_3$-$H_2O$ nucleation alone (Kirkby et al., 2011). Sihto et al. (2006), Kuang et al. (2008) and Paasonen et al. (2010) developed parameterisations based on ground based observations of boundary layer nucleation events. These parameterisations are typically least square fits to a power law dependency of observed particle formation rates as a function of vapour concentration and are only valid for environments that match the observation sites.

New parameterisations of aerosol nucleation based on experiments in the CERN CLOUD chamber were published in the past years. These parameterisations include a variety of chemical species and in most cases the influence of air ions. Additionally, these parameterisations offer a description of boundary layer and upper tropospheric nucleation. Dunne et al. (2016) derived parameterisations for systems that include $H_2SO_4$, $NH_3$ and ions over a wider range of atmospheric temperatures. Riccobono et al. (2014) describes secondary organic aerosol nucleation from biogenic vapours and $H_2SO_4$, while Kirkby et al. (2016)

showed that nucleation can even occur without $H_2SO_4$, purely from biogenic vapours and air ions. Furthermore, Riccobono et al. (2014) and Kirkby et al. (2016) provided a parameterisation used by Gordon et al. (2016) to study the effect of NPF on climate. Most of the recent parameterisations of particle formation use atmospheric ions or ionising radiation (Yu, 2010; Kazil et al., 2010; Dunne et al., 2016; Kirkby et al., 2016).

Aside from production of aerosol particles the chemical conversion and transport of aerosols in the atmosphere are of im-

portance. Various General Circulation Models (GCM) include aerosols to study global aspects of aerosol particles. Mann et al. (2014) compared 12 global Chemical Transport Models (CTM) and GCM, which included aerosol micro-physics. Estimates on the fraction of CCN particles from secondary aerosol formation varies between different models, e.g. Merikanto et al. (2009); Yu and Luo (2014).

In this work the implementation of the CLOUD based parameterisations into the Modular Earth Submodel System (MESSy)

is described, as well as their application in the EMAC chemistry-GCM. These parameterisations are part of the New Aerosol Nucleation submodel (NAN). The new parameterisation requires the inclusion of tropospheric and stratospheric ions, therefore the submodel IONS treating production of ions from galactic cosmic rays and radon was created.

## 2 Methods

### 2.1 MESSy

MESSy is a collection of models for various aspects of Earth system modelling. Most of the models are organised as submodels, which form the submodel core layer (SMCL). Models in the SMCL can either be used as box model or be part of a larger model, the so called base model. A commonly used combination of MESSy with a GCM is EMAC (Pozzer et al., 2012; Klingmüller et al., 2014). Initialisation and acquiring data from other submodels is done within the submodel interface layer (SMIL). The





control of each submodel is performed through variables in Fortran 90 namelists. Each submodel uses a file with these namelists to set variables and allow coupling to other submodels. As described in Jöckel et al. (2010), submodels can share values via the channel infrastructure.

Several submodels describing aerosol dynamics exist within the MESSy framework. The current most-developed submodels for aerosol dynamics within the MESSy framework are GMXe (Pringle et al., 2010), MADE and its successor MADE3 (Lauer et al., 2005). The GMXe submodel is based on M7 (Vignati et al., 2004), which describes the aerosol size distribution as seven overlapping log-normal distributions, of which 4 modes are soluble and 3 modes are insoluble. M7 and GMXe were developed and optimised for inorganic aerosol particles, therefore Tsimpidi et al. (2014) developed the ORACLE submodel for the treatment of Secondary Organic Aerosols (SOA), see also Tsimpidi et al. (2017). ORACLE uses the volatility basis set approach based on Donahue et al. (2006) to calculate partitioning of gases between the particle and gas phases. The aerosol particle size distribution is taken from GMXe. Gas phase chemical reactions are calculated with the MECCA submodel (Sander et al., 2011).

## 2.2 IONS submodel

Atmospheric ions are produced by galactic cosmic rays (GCRs) and by the radioactive decay of radon and its subsequent decay products. In order to provide ion pair production rates independent of the GEC submodel (Baumgaertner et al., 2013), ion pair production and the calculation of a steady state ion concentration were included in a new MESSy submodel IONS. For the calculation of ion pair production from Radon decay the DRADON submodel (Jöckel et al., 2010) must provide tendencies for all tracers in the decay chain. The submodel can provide the ion pair production rate and steady state ion pair concentration to other submodels via MESSy's coupling scheme.

Radon emissions are described either by constant emissions over land (value set via namelist) and ocean (also set via namelist), or by an emission flux map, e.g. Zhang et al. (2011). For a detailed description of possible input parameters see the electronic supplement. The ion pair production from a single decay event is calculated in the same way as described by Zhang et al. (2011). It is assumed that each $\alpha$ decay creates an ion pair for every 35.6 eV of initial energy, while every $\beta$ decay produces an ion pair for every 32.5 eV of initial energy. The radon decay chain and the corresponding energies are given by the reaction chain given in R1 to R5. Half life times are given above the reaction arrows.

$$^{222}_{86}\text{Rn} \xrightarrow{3.8\text{d}} {}^{218}_{84}\text{Po} + \alpha \ 5.59\text{MeV} \tag{R1}$$

$$^{218}_{84}\text{Po} \xrightarrow{180\text{s}} {}^{214}_{82}\text{Pb} + \alpha \ 6.12\text{MeV} \tag{R2}$$

$$^{214}_{82}\text{Pb} \xrightarrow{27\text{min}} {}^{214}_{81}\text{Bi} + \beta \ 1.02\text{MeV} \tag{R3}$$

$$^{214}_{81}\text{Bi} \xrightarrow{20\text{min}} {}^{210}_{82}\text{Pb} + \beta + \alpha \ (7.88 + 3.27)\text{MeV} \tag{R4}$$





$$_{82}^{210}\text{Pb} \rightarrow ... \xrightarrow{22.3\text{y}} {}_{82}^{206}\text{Pb} + \alpha \tag{R5}$$

The $\alpha$-decay of $^{214}$Po to $^{210}$Pb is not explicitly mentioned in R4 due to a half life time of only 164 $\mu$s, though the released $\alpha$ particle is included in the calculation of produced ion pairs. The radon decay chain ends with the stable isotope $^{206}$Pb. Under atmospheric conditions however, if the optional coupling of DRADON submodel to an aerosol model is chosen, $^{210}$Pb is already taken up into aerosol particles, due to a lifetime with respect to radioactive decay of 22.3 years. Since the half life time of this decay exceeds the lifetime of atmospheric aerosols by more than two orders of magnitude the last decay chain is not included in the model.

The IONS submodel includes the Cosmic Ray Induced Ionisation (CRII) scheme by Usoskin et al. (2010). The CRII tables contain the ion pairs produced per second and gram of air as function of atmospheric depth, cosmic ray modulation and geomagnetic cut off rigidity. Values between the tabulated points are calculated by linear interpolation in the same way as in Dunne et al. (2016). The geomagnetic cut off rigidity is calculated by the method of Fraser-Smith (1987). The main difference between this implementation and the one described in Dunne et al. (2016) is the use of more recent tables for both the modulation of GCRs and and geomagnetic cut off rigidity. For the GCR modulation a choice between a table of monthly averages from 1936-2016 (Usoskin et al., 2005; McCracken and Beer, 2007; Usoskin et al., 2011) or yearly averages since 1600 (Asvestari and Usoskin, 2016; Asvestari et al., 2017) is available. The MESSy import for time series data provides a linear interpolation for dates between the listed values. The geomagnetic cut off rigidity uses the first 3 coefficients of the IGRF coefficients of Earth's magnetic field. For 1900-2015 the IGRF table is applied, while for years prior to that the reconstruction of the magnetic field by Jackson et al. (2000) is used. The coupling to the GEC model makes it possible to use the new parameterisation of ionisation in the GEC submodel to calculate the conductivity of air.

The number concentration of small ion pairs $n^\pm$ due to production and their loss in the atmosphere can be described by

$$\partial_t n^\pm = Q_d + Q_g - k_r n_\pm^2 - k_a A n_\pm - J_i. \tag{1}$$

The first two terms $Q_d$ and $Q_g$ are the ion pair production due to radioactive decay and galactic cosmic rays. The other terms describe the various loss processes. The first loss process is ion-ion recombination. The rate constant of ion-ion recombination $k_r$ is calculated with the parameterisation of Brasseur and Chatel (1983) which gave reasonable agreement with ion-ion recombination in the CERN CLOUD chamber (Franchin et al., 2015), although under high-pressure, low-temperature conditions. The second loss process is uptake of ions by aerosol particles with a number concentration of $A$. The particle size dependent coefficient $k_a$ is calculated using the same method as in Tinsley and Zhou (2006) and Baumgaertner et al. (2013). For particles with a radius larger than 10 nm, the expression

$$k_a = 4.36 \cdot 10^{-5} r_{\mu m} - 9.2 \cdot 10^{-8} \tag{2}$$

from Hoppel (1985) is used to calculate the attachment rate coefficient. $r_{\mu m}$ is the aerosol particle diameter in $\mu$m. For particles smaller than this radius Tinsley and Zhou (2006) provided,

$$\log_{10} k_a = 1.243 \log_{10} r_{\mu m} - 3.978 \tag{3}$$



as extrapolation for nucleation mode particles. The radius of the aerosol particles is provide by aerosol submodels such as GMXe.

The third loss process is ion-induced nucleation, which is negligible outside of nucleation events but becomes important during nucleation events. However, this loss is only taken into account in the nucleation submodel when calculating the ion-induced nucleation rate. The reason for this is to limit the maximum possible ion induced nucleation to the ion pair production rate. Nevertheless, small ions that are lost due to nucleation simply become slightly larger ions and removing them from the simulation can cause an inbalance in the small ion concentration. Since only small ions are considered here this would lead to an overall ion inbalance. Furthermore, singly charged particles up to a diameter of a few nm have the same recombination coefficient, see for example Hoppel (1985) or López-Yglesias and Flagan (2013). Therefore, losses due to nucleation are not used in the ion submodel.

## 2.3 NAN submodel

The channel objects in MESSy (Jöckel et al., 2010) allow for a flexible transfer of variables between models. Therefore, the implementation of the aerosol nucleation parameterisations can be used by several aerosol submodels within MESSy. Further, this approach allows code which is easier to maintain and adjust to new scientific findings, such as refined parameterisations, including additional species and new nucleation mechanisms. The steady-state new particle formation rates described in Dunne et al. (2016), Kirkby et al. (2016) and Riccobono et al. (2014) were implemented into the nucleation model core layer of the submodel NAN (New Aerosols Nucleation). This results in several functions that return the formation rates of aerosol particles with a diameter of 1.7 nm. A short summary of the parameterisation will be given here, while details, such as the choice of functions, number of parameters and optimisation are explained in the supporting information of Dunne et al. (2016), Kirkby et al. (2016) and Riccobono et al. (2014). The neutral binary homogeneous nucleation of sulphuric acid and water is given by

$$J_{b,n} = k_{b,n}(T) \left[\text{H}_2\text{SO}_4\right]^{p_{b,n}} \tag{4}$$

and neutral homogeneous ternary nucleation of sulphuric acid, ammonia and water by

$$J_{t,n} = k_{t,n}(T) f_n \left([\text{H}_2\text{SO}_4], [\text{NH}_3]\right). \tag{5}$$

The indices indicate the type of nucleation with $b$ binary, $t$ ternary, $n$ neutral and $i$ ion induced nucleation. The function $k_{x,y}(T)$ has the same form for all four nucleation pathways but uses different parameters and basically describes the temperature dependence of the particle formation rate as

$$\ln k_{x,y}(T) = u_{x,y} - \exp\left(v_{x,y}\left(\frac{T}{1000\text{K}}\right) - w_{x,y}\right), \tag{6}$$

with $x \in (b,t)$, $y \in (n,i)$ and the temperature $T$ in K. The function

$$f_y\left([\text{H}_2\text{SO}_4], [\text{NH}_3]\right) = \frac{[\text{H}_2\text{SO}_4]^{p_{t,y}} [\text{NH}_3]}{a_y + \frac{[\text{H}_2\text{SO}_4]^{p_{t,y}}}{[\text{NH}_3]^{p_{a,y}}}} \tag{7}$$



is shared with the ion-induced ternary channel and controls the saturation behaviour of the ternary nucleation. The equations for ion induced nucleation take a similar form but with the concentration of negative ions, $[n^-]$, included as a factor. This leads to

$$J_{b,i} = k_{b,i}(T) [n^-][\text{H}_2\text{SO}_4]^{p_{b,i}} \tag{8}$$

and

$$J_{t,i} = k_{t,i}(T) [n^-] f_i([\text{H}_2\text{SO}_4],[\text{NH}_3]). \tag{9}$$

Although called binary and ternary nucleation, the influence of water vapour is not explicitly indicated in the parameterisation. Although the experimental data that forms the basis of this parameterisation was conducted at various water vapour concentrations, most of the measurements were done at a relative humidity of 38%. Dunne et al. (2016) give a scaling factor dependent

on the relative humidity as fraction, $RH$, and temperature in, $T$, in Kelvin

$$f_{RH} = 1 + c_1 (RH - 0.38) + c_2 (RH - 0.38)^3 (T - 208\text{K})^2, \tag{10}$$

with $c_1 = 1.5$ and $c_2 = 0.045\ \text{K}^{-2}$. However, this scaling factor is more of an ad hoc solution and based oni very few measurements. The overall effect of this scaling is described as relatively small in Dunne et al. (2016) and is not used here.

Two functions describe nucleation by oxidised organic species, named HOM in Kirkby et al. (2016), which is again split

into a neutral channel

$$J_{K,n} = a_1 [\text{HOM}]^{a_2 + \frac{a_5}{[\text{HOM}]}} \tag{11}$$

and an ion induced channel

$$J_{K,i} = \left([n^-] + [n^+]\right) a_3 [\text{HOM}]^{a_4 + \frac{a_5}{[\text{HOM}]}}. \tag{12}$$

A major difference between this channel and equation 8 or 9 is that the organic nucleation can proceed with positive and

negative ions. The original form of eq. 12 given by Kirkby et al. (2016) assumed charge balance; the equation given above remains valid even if charge balance is not given.

The description of nucleation from oxidised organic species and sulphuric acid is described according to the power law dependency of Riccobono et al. (2014). The definition of oxidised organic species varies between Kirkby et al. (2016) and Riccobono et al. (2014). The latter defined the oxidised organics as BioOxOrg, which are produced by the oxidation of pinanediol

with OH radicals, while the former named the oxidised organics HOM and defined it as a product of $\alpha$-pinene oxidation by O$_3$ and OH. Mass spectra from both sets of experiments show similar species with high oxygen to carbon ratios, so it can be assumed that the nucleating species are also the same to a large extent. However, Riccobono et al. (2014) only provides evidence for nucleation of OH oxidation products with sulphuric acid. While it is reasonable to assume that O$_3$ oxidation products will also nucleate with H$_2$SO$_4$, the parameterisation is strictly only valid for OH oxidation products. An additional problem is that

the nucleation rate parameterisation given in Kirkby et al. (2016) cannot be separated into nucleation channels driven by OH





and $O_3$ oxidation products. Therefore, it is assumed that the species HOM is the sum of monoterpene oxidation products from $O_3$, denoted $HOM_{O_3}$ and OH radicals, $HOM_{OH}$. With this definition the power law dependence from Riccobono et al. (2014) can be writen as

$$J_R = k_R[\text{H}_2\text{SO}_4]^2[\text{HOM}_{\text{OH}}]. \tag{13}$$

The yield of $HOM_{OH}$ production, 0.6% for lumped atmospheric terpenes according Tröstl et al. (2016), was included in the parameter $k_R$ since the original parameterisation did not include a yield.

Nucleation between amines and sulphuric acid is described as

$$J_A = k_{A,1}[\text{Amines}]^{p_{a,1}}[\text{H}_2\text{SO}_4]^{p_{s,1}}, \tag{14}$$

if $[\text{Amines}] > 2.0 \cdot 10^8 \text{cm}^{-3}$ and

$$J_A = k_{A,2}[\text{Amines}]^{p_{a,2}}[\text{H}_2\text{SO}_4]^{p_{s,2}} \tag{15}$$

in all other cases. This approach is the same as in Dunne et al. (2016), with a a more generalised notation of the parameters. This allows straightforward and flexible switching between different parameterisation for amine nucleation. This is also of importance since different amine species can have different nucleating potential (Jen et al., 2014; Glasoe et al., 2015). The parameterisation of Bergman et al. (2015) can easily be applied by setting the threshold concentration to 0 and setting the

parameters with integer index 1 to the values used in Bergman et al. (2015).

With all nucleation pathways, $J_j$, described above the total nucleation rate is described as

$$J_{total} = \sum_j J_j \,, \tag{16}$$

the sum of all particle formation rates. It is assumed here that the different nucleation channels do not interact with each other as subcritical clusters or particles below the threshold of 1.7 nm.

All fit parameters can be set in the nucleation submodels namelist `PARAM` (see electronic supplement for details). If no setting is chosen the published default values are used. This makes it possible to study the sensitivity of model results to these parameters and change parameterisations easily. None of the organic nucleation channels described above have an experimental basis for a temperature dependence of the nucleation rate. Nevertheless, a temperature dependence is defined in the model using an exponential scaling factor,

$$\gamma = \exp(BT), \tag{17}$$

which is applied to Eq. 11, 12 and 13. Setting the parameter $B = 0$ leads to no temperature dependence in the model for the organic nucleation channels and is the default setting.

The existing subroutines for calculating nucleation rates according to the parameterisations of Vehkamäki et al. (2002) and Kulmala et al. (1998) were copied from GMXe so that these legacy nucleation parameterisations can also be used. The set





of parameterisations for a model run is set in the submodels namelist. If multicomponent nucleation is chosen the submodel tests whether nucleation depletes the gas-phase concentration of nucleating vapours. If this is the case, an Euler integration is performed for the length of the global model time step which calculates the vapour depletion, derives the average particle formation rate for each pathway and the total number concentration of newly formed particles.

The newly formed particles can either be added directly to the nucleation mode, as is done in GMXe, or optionally the method of Anttila et al. (2010) can be used to grow the freshly formed particles to a fixed size. The latter method is useful if the smallest size bin or mode of the aerosol model is larger than the size of the nucleated particles. The implementation of Anttila et al. (2010) into MESSy does not include iteration, in order to keep computational cost at a minimum. The condensation sink is provided by the aerosol dynamics model via MESSy's channel objects. The major drawback of this approach is that

it requires additional parameterisations for the growth rates of freshly nucleated aerosol particles. For use with GMXe, the freshly nucleated particles are added directly into the nucleation mode.

### 2.4 Simulations

Nucleation rates in MESSy are usually calculated within the calling aerosol submodel. Therefore EMAC simulations were performed to evaluate whether the call to the nucleation subroutine can be moved outside of GMXe. A simulation that used the

Dunne et al. (2016) parameterisation within the GMXe submodel served as baseline for comparison with the new nucleation submodel. This baseline simulation was compared with a simulation where the new submodel was called after GMXe and a simulation with the nucleation called before GMXe.

    A full list of the simulations is given in Table 1. The set of chemical reactions in these simulations was the same as in Jöckel et al. (2016). Simulations were carried out with a spectral resolution of T42 and 31 hybrid-$\sigma$-pressure levels. The dynamics

were nudged towards ERA-interim data of the European Centre of Medium range Weather Forecast (ECMWF). Tracer nudging and data initialisation were the same as in Jöckel et al. (2016).

    To test the organic nucleation scheme, the years 2007 and 2008 were simulated, with the first year acting as spin up. The chemical reactions and emissions from Tsimpidi et al. (2014) were used. Reactions of terpenes with OH and ozone that form HOM species were added, similar to Gordon et al. (2016) with the refined yields from Tröstl et al. (2016). As mentioned in

section 2.3, the terpene oxidation product is split into the product of ozonolysis of terpenes and oxidation of terpenes with OH radicals, leading to

$$\mathrm{LTERP} + \mathrm{O_3} \rightarrow \mathrm{HOM_{O_3}} \tag{R6}$$

and

$$\mathrm{LTERP} + \mathrm{OH} \rightarrow \mathrm{HOM_{OH}} \tag{R7}$$

as the reactions of the aerosol precursor gas. The lumped terpene tracer, LTERP, is based on terpene emissions from Tsimpidi et al. (2014). The gas to particle phase partitioning of the added organic species is calculated by ORACLE (Tsimpidi et al., 2014). A saturation vapour pressure of $2 \cdot 10^{-2}$ µgm$^{-3}$ was assumed for HOMO$_\mathrm{OH}$ and HOMO$_\mathrm{O_3}$. This places the saturation vapour pressure within the LVOC regime as described in Tröstl et al. (2016).





**Table 1.** Overview of the EMAC simulations. The chemistry was taken from Jöckel et al. (2016). The column *experiment* gives the name of the experiment, which is used for axis labelling in figures. Column *parameterisation* gives the citation for the nucleation parameterisation used. *Position* in EMAC indicates in which part of the code the nucleation rate was calculated.

| Experiment | Parameterisation | NAN called |
|---|---|---|
| GMXe | Dunne et al. (2016) | in GMXe |
| Dunne 1 | Dunne et al. (2016) | before GMXe |
| Dunne 2 | Dunne et al. (2016) | after GMXe |
| Organic | Riccobono et al. (2014); Dunne et al. (2016); Kirkby et al. (2016) | after GMXe |

The SCOUT submodel provides instantaneous values of nucleation rates, aerosol particle and precursor gas concentrations at each 600 s model time step at the coordinates of 22 atmospheric measurement stations from the EBAS database (Tørseth et al., 2012). The stations and their coordinates are given in Tab. 2. The year 2008 was chosen for the overlap with ion measurements from Manninen et al. (2010). The aerosol particle number concentrations were measured with condensation particle counters, which provide the total concentration of particles exceeding a threshold diameter. For comparison with observational data, the concentration of particles $N_d$ exceeding a diameter $d$, here 10 nm, is calculated as

$$N_d = \sum_{j=1}^{m} N_j \left( 1 - \mathrm{erf} \left( \frac{\ln \left( d/D_j^p \right)}{\sqrt{2} \ln \sigma_j} \right) \right), \tag{18}$$

for a set of $m$ modes, in the case of GMXe $m = 7$, of overlapping log normal size distributions. The count mean diameter for mode $j$ is given by $D_j^p$ and the standard deviation as $\sigma_j$.

# 3 Results

## 3.1 Ion model evaluation

Six of the 22 stations listed in EBAS with aerosol particle data for 2008 (see table 2), were used in the analysis of ion spectrometer measurements in Manninen et al. (2010). The ion concentration measured at these stations is compared to the simulated concentration in Fig. 1. For this plot the measured concentration of positive and negative ions was averaged in order to compare with the simulation, which retains ion balance. The simulated time series was matched onto the observed time series by linear interpolation, using the timestamps of the observation as grid for both time series. Simulation and observation are in good agreement for most data points, with 65% of the data points within a factor of 2 and 93% within a factor of 5. However, EMAC also tends to over predict ion concentrations by a factor of up to 2 in many cases, typically when the observed ion concentration is below 500 i.p. cm$^{-3}$. This can in part be attributed to model assumptions, e.g. ion balance and the lack of a binned ionised aerosol model, and in part to the instruments used for the measurements. Wagner et al. (2016) showed that the





transmission efficiency for NAIS/AIS can be as low as 70% for small ions, depending on instrument and inversion used. This correction cannot be applied ad hoc to historic measurements due to changes in instruments and inversions. Nevertheless, this provides an indication that the measured small ion concentrations may be too low by up to a factor of approximately 0.7.

Certain specific events in high altitude locations which can lead to high ion concentrations, such as splashing rain drops (Tammet et al., 2009) or strong wind episodes (Virkkula et al., 2007), are not accounted for in the model. These events are the reason why the plot was limited to 3000 i.p. cm$^{-3}$ on both axes as some observations showed extremely high ion concentrations for certain days. All the observed ion concentrations exceeding 3000 cm$^{-3}$ in Fig. 1 were measured at the high-altitude stations.

Time series and distributions of monthly ion concentrations, modelled and measured for 2 stations (Hyytiälä and Hohenpeissenberg), are shown in Fig. 2. The blue (left) part of each area shows the distribution of simulated small ion concentrations, while the red part (right) shows the measured concentrations. The horizontal dashes in each area give the quantiles. The distribution for the high elevation site Hohenpeissenberg shows a few extremely high ion concentrations of up to 6000 i.p. cm$^{-3}$. These are common on high elevation sites and Manninen et al. (2010) attributed their formation to strong winds. The low level station at Hyytiälä shows no such behaviour. The time series indicate also that the model does not capture the seasonality shown in the observations. This can have various reasons, such as seasonality in the Radon emissions or differences in the aerosol number concentrations and hence differences in losses of ions to aerosol particles between model and observation. However, the data set shown here is rather small and lacks some measurements in the first months of 2008.

Figure 3 shows the zonal distribution of the total ion pair production rate for the year 2008. Ion pair production rates are highest close to the poles and at pressure levels of around 200 hPa, due to higher flux of GCR particles close to the magnetic poles. The ion pair production rate is a factor of 2 lower along the equator at these pressure levels. Towards ground level the effect of GCR particles becomes less important and radon decay becomes an important contributor over land. Fig. 4 shows the global ion pair production rates at ground level(upper panel) and at 200 hPa (lower panel). The ground level distribution shows that ion production over land exceeds the production over oceans. This is due to radon emissions over land. Examining the production rate over the oceans shows a negligible dependence on the latitude. At 200 hPa the latitude correlates with the ion pair production due to Earth's magnetic field. The orientation of the magnetic field also causes the sinusoidal shape visible in the distribution. The overall distributions of small airborne ions and ion pair production rates obtained with EMAC agree well with similar simulations from other models, e.g. Usoskin et al. (2008) and Baumgaertner et al. (2013).

### 3.2 Nucleation Model evaluation

#### 3.2.1 Intramodel comparison

Comparison between the new implementation of the Dunne et al. (2016) parameterisation outside the GMXe submodel and an implementation within GMXe is done by comparing number concentrations for all soluble modes in all grid cells at 10 h intervals over a given month. Fig. 5 shows the aerosol particle number concentration from the EMAC simulations. The ordinate axis shows values with the nucleation calculated within GMXe, while the abscissa axis shows values of particle formation rates calculated in NAN before the call to GMXe. The panels show the results for the two smallest soluble modes, nucleation and





Aitken mode, in GMXe. The color indicates the total number of occurrences within each hexagonal bin. Most values differ by less then a factor of 10, indicated by the dashed lines. The percentage of points within a factor of 2, 5 and 10 are 84%, 94% and 96% respectively.

Figure 6 is the same as Fig. 5, except that the NPF rate was calculated after GMXe calculated the aerosol size distribution for the time step. Calling the nucleation submodel after GMXe gives slightly better agreement with the baseline model, with 88% of points within a factor of 2. The difference between the implementation before and after GMXe is the result of numerical errors due to the linearisation of non-linear processes. Similar effects can be expected for other submodels within MESSy. To test this, the GMXe submodel was called with the radiation microphysics or with general physics and the difference between these two simulations leads to a comparable statistics as the presented comparison between GMXe and NAN.

### 3.2.2 Comparison with observations

A comparison between atmospheric observations and modelled particle concentrations, for 22 locations from the EBAS (Tørseth et al., 2012) database, are shown in Figure 7. For the comparison with observations, a cut-off diameter of 10 nm was used since most CPCs in the database appear to exceed a 50% counting efficiency at this size. The simulated time series of particle concentrations was matched onto the observed time series by linear interpolation, using the timestamps of the observation as grid for both time series. The overall agreement between both data sets is good, 44% of the data within a factor of 2, 77% within a factor of 5 and 88% within a factor of 10. However, it is clear that the difference between both data sets is not normal distributed. Excellent agreement exists in a large central area of the distribution.

For two of the stations, the monthly distributions of particle concentrations are shown in Fig. 8. The left (blue) part of the areas give the distribution from the EMAC simulation for each month, while the right (red) areas are from observations. The central horizontal line indicates the median concentration, the upper and lower vertical lines the 1st and 3rd quantiles. The missing right areas for Bondville indicate missing data. From this plot it can be seen that EMAC and observations differ to varying degrees in their distribution of values within each month. Nevertheless the model catches certain seasonality for some stations, shown here for Hyytiälä, while the seasonality predicted by EMAC is not evident from the observational data for Bondville. This can best be seen from the medians. Additionally, the observations go through certain extreme values which are in most cases not exceeded by the model (aside from two months in Hyytiälä). This could be due to not yet included nucleation mechanisms or local pollution events not captured by a global model.

## 4 Conclusions

Two new submodels were introduced to MESSy and tested with EMAC. The submodel IONS provides ion pair production rates that can be used in other submodels such as GEC (Baumgaertner et al., 2013) or the here presented NAN submodel. NAN calculates new particle formation rates based on several optional nucleation parameterisations. Having the nucleation rates outside of the aerosol microphysic models comes with several advantages. New parameterisations can be implemented




easily without major rearrangements in existing source code. The same parameterisations can be used by different aerosol microphysical models. Furthermore, the submodel can be used in a box model or other base models.

The calculated ground-level ion concentration was compared to a small set of field measurements and overall gives reasonable agreement. Some extreme events are not reproduced by the model, perhaps due to a lack of suitable parameterisations,
unknown microphysical process or their potentially localised nature. The global distribution of ion-pair production rates follows known patterns from theoretical considerations and numerical models.

The effect of calculating nucleation rates outside of GMXe has some influence on the results. This is expected when linearising non linear processes and is an intrinsic problem of operator splitting. Nevertheless, it has been shown that the new submodel NAN agrees well with results from GMXe, with 84% of the data within a factor of 2.
Large uncertainties remain, mainly due to the incomplete nature of the implemented nucleation rate parameterisations. Incomplete aspects include the temperature dependence of nucleation involving organic species, the chemistry of HOM formation and details about the interaction of the parameterisations of Riccobono et al. (2014) and Kirkby et al. (2016). The latter is in part due to the different definition of oxidised organic species, to different instrumentation available, and to differences in the experimental design. The largest open question is certainly whether the parameterisation in Riccobono et al. (2014) is also
valid for species from terpene ozonolysis.

**Code availability**

The Modular Earth Submodel System (MESSy) is continuously further developed and applied by a consortium of institutions. The usage of MESSy and access to the source code is licensed to all affiliates of institutions which are members of the MESSy Consortium. Institutions can become a member of the MESSy Consortium by signing the MESSy Memorandum of
Understanding. More information can be found on the MESSy Consortium Website (http://www.messy-interface.org). The code presented here has been based on MESSy version 2.53.0 and will be available in version 2.54.0.

*Competing interests.* The authors declare that they have no conflict of interest.

*Acknowledgements.* The service charges for this open access publication have been covered by the Max Planck Society.



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





**Table 2.** Measurement stations used in the comparison with atmospheric particle concentrations in Fig. 8 and 7. Station coordinates taken from the EBAS data files. *Station names in Italic indicate locations with ion measurements.*

| Station | Lat | Lon | altitude | Environment |
|---|---|---|---|---|
| Barrow | 71.32 | -156.61 | 11 | remote, polar, marine |
| Bondville | 40.05 | -88.37 | 213 | rural |
| Cape Point | -34.35 | 18.49 | 230 | marine, rural |
| Cape San Juan | 18.381 | -65.62 | 65 | marine, rural |
| Finokalia | 35.32 | 25.67 | 250 | marine, remote |
| Gosan | 33.28 | 126.17 | 89 | marine, rural |
| Harwell | 51.57 | -1.32 | 126 | rural |
| *Hohenpeissenberg* | 47.80 | 11.01 | 988 | rural |
| *Hyytiälä* | 61.85 | 24.28 | 181 | rural |
| Izana | 28.31 | -16.50 | 2373 | high-altitude, marine, remote |
| *Jungfraujoch* | 46.55 | 7.99 | 3580 | high-altitude, remote |
| Lulin | 23.47 | 120.87 | 2862 | high-altitude, rural |
| *Mace Head* | 53.33 | -9.90 | 10 | marine, remote |
| Mt Cimone | 44.18 | 10.70 | 2165 | high-altitude, remote |
| Neumayer | -70.67 | -8.27 | 42 | remote, polar, marine |
| *Pallas* | 67.97 | 24.12 | 565 | remote, polar |
| Preila | 55.38 | 21.03 | 5 | marine, rural |
| *Puy de Dome* | 45.77 | 2.95 | 1465 | high-altitude, rural |
| Samoa | -14.25 | -170.56 | 77 | marine |
| Southern Great Plains | 36.6 | -97.5 | 300 | rural |
| Steamboat Springs | 40.45 | -106.74 | 3220 | high-altitude |
| Trinidad Head | 41.05 | -124.15 | 107 | marine, rural |

Yu, F. and Luo, G.: Effect of solar variations on particle formation and cloud condensation nuclei, Environmental Research Letters, 9, 045 004, doi:10.1088/1748-9326/9/4/045004, http://stacks.iop.org/1748-9326/9/i=4/a=045004, 2014.

Zhang, K., Feichter, J., Kazil, J., Wan, H., Zhuo, W., Griffiths, A. D., Sartorius, H., Zahorowski, W., Ramonet, M., Schmidt, M., Yver, C., Neubert, R. E. M., and Brunke, E.-G.: Radon activity in the lower troposphere and its impact on ionization rate: a global estimate using different radon emissions, Atmospheric Chemistry and Physics, 11, 7817–7838, doi:10.5194/acp-11-7817-2011, http://www.atmos-chem-phys.net/11/7817/2011/, 2011.





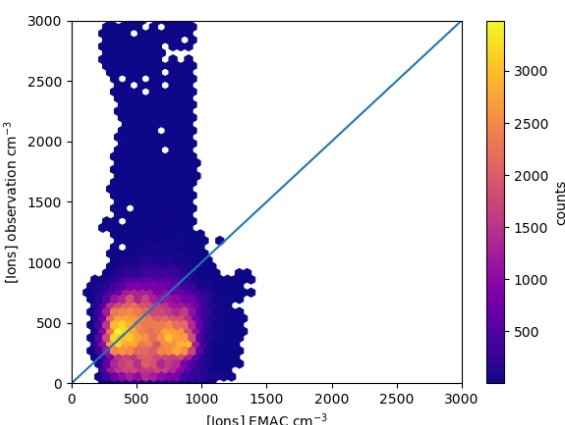

**Figure 1.** Comparison of observed ground level ion concentration with simulated concentration at the six measurement sites with ion measurements (Tab. 2).





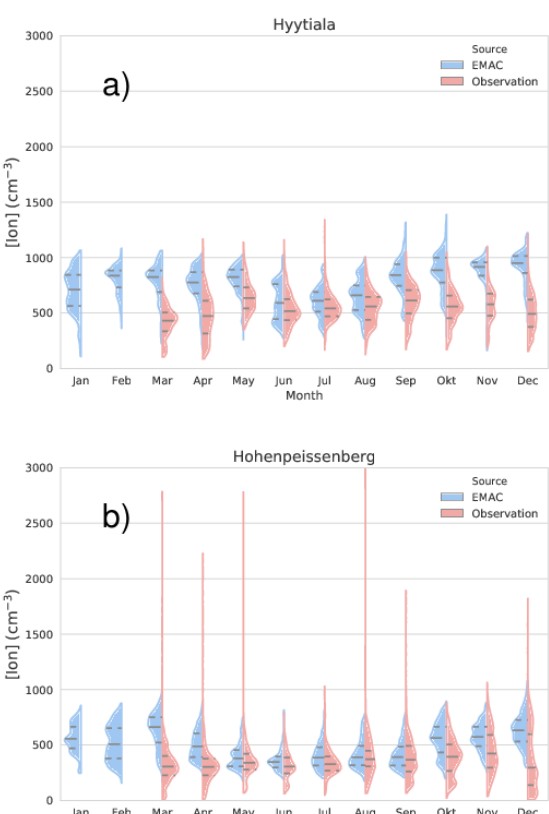

**Figure 2.** Monthly distribution of observed and simulated ion concentration at two locations in 2008. The station codes are above each panel. Blue areas (left half of each area) modelled distribution, red (right) areas observed values. For January and February 2008 no station had ion data available.





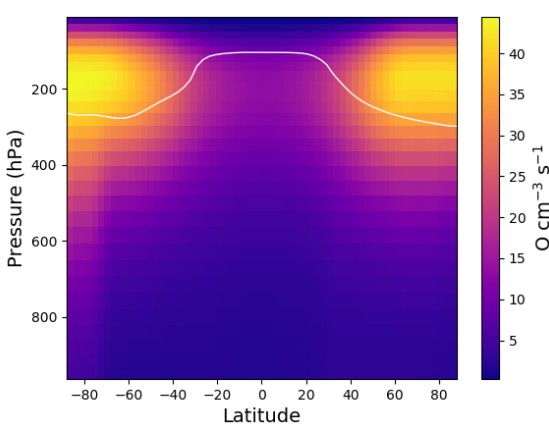

**Figure 3.** Zonal average yearly mean ion pair production rate, $Q$, from EMAC for 2008. The white line shows the tropopause.





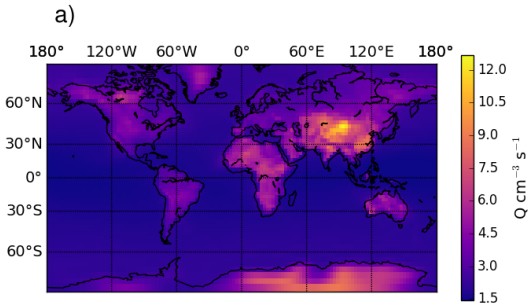

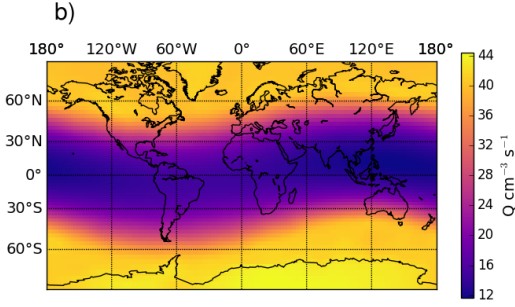

**Figure 4.** Global distribution of yearly mean ion pair production rate, $Q$, at ground level (upper panel) and at 200 hPa (lower panel).




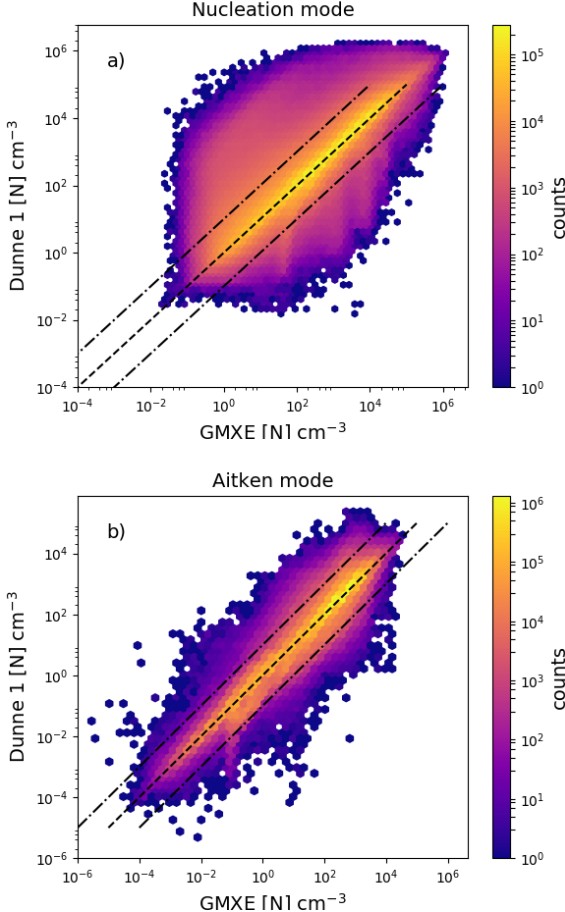

**Figure 5.** Logarithm of the aerosol particle number concentration with the Dunne et al. (2016) nucleation scheme implemented in NAN and called in EMAC just before the call of GMXe (y-axis) vs implementation inside the GMXe submodel (x-axis). Panel a) shows the results for nucleation mode particle and panel b) for aitken mode particles. The colour indicates total number of counts in each hexagonal bin.





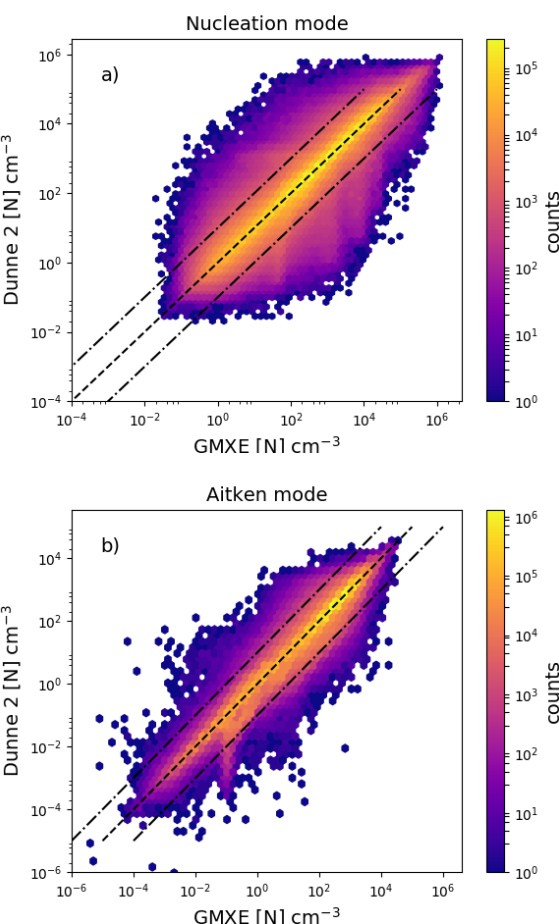

**Figure 6.** Logarithm of the aerosol particle number concentration with the Dunne et al. (2016) nucleation scheme implemented in NAN and called in EMAC just before the call of GMXe (y-axis) vs implementation inside the GMXe submodel (x-axis). Panel a) shows the results for nucleation mode particle and panel b) for aitken mode particles. The colour indicates total number of counts in each hexagonal bin.





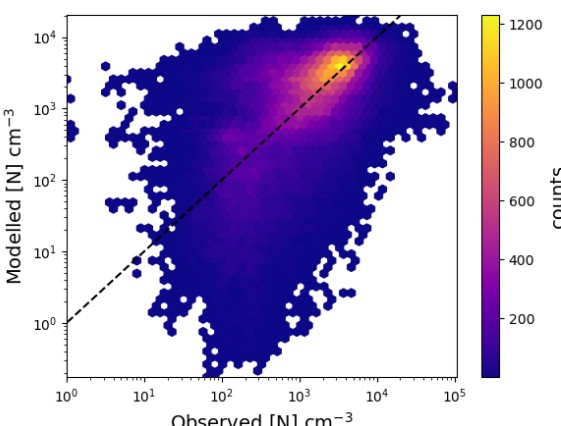

**Figure 7.** Comparison of particle concentration from EMAC with atmospheric observations for the year 2008. The used stations and their coordinates are in table 2.





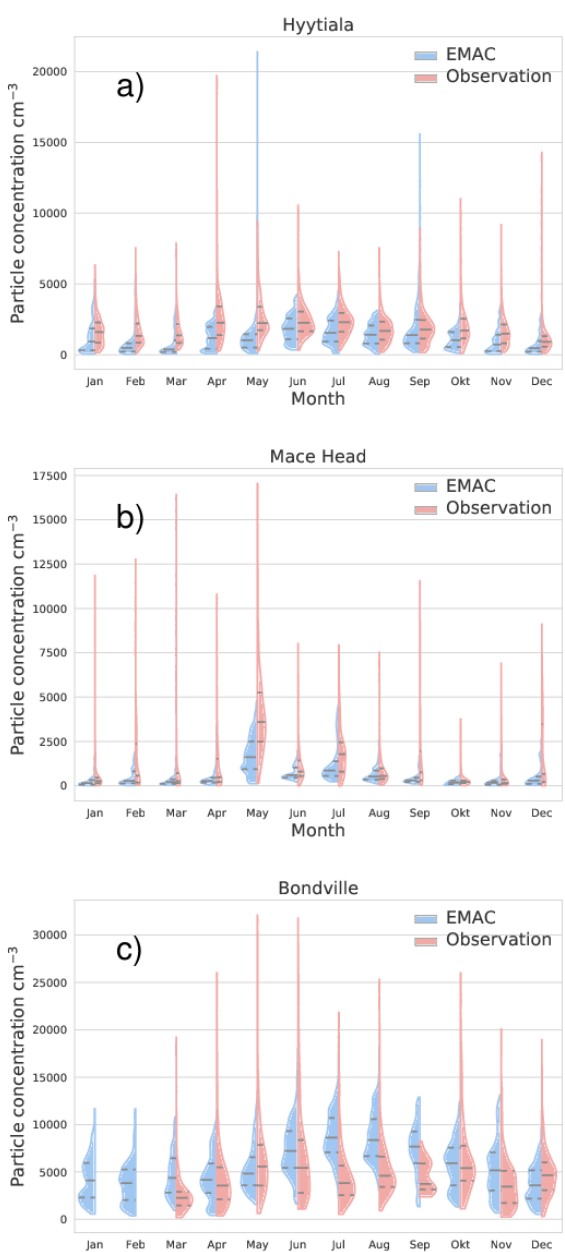

**Figure 8.** Comparison of EMAC simulated aerosol particle number concentrations, including the parameterisations of Riccobono et al. (2014),Dunne et al. (2016) and Kirkby et al. (2016), with atmospheric observations for 3 stations and the year 2008. The area shows an estimate of the monthly distribution of values for EMAC simulation (left, blue) and observation (right, red). The central vertical line within each area gives the monthly median for each month, the upper and lower the lines are 1st and 3rd quantiles. The station names are above each panel, table 2 contains the coordinates for each station. Figures for all stations are in the supplementary.