# Peer review of "Two new submodels for the Modular Earth Submodel System (MESSy): New Aerosol Nucleation (NAN) and small ions (IONS) version 1.0"

_Geoscientific Model Development, 2018_

## Referee Comment (RC1) · Anonymous Referee #1 · 6 Sep 2018

This paper documents the creation of two modules for use for the MESSy system. It does fine for providing this documentation, and my comments are relatively minor.

P2 L4: Merikanto et al. (2007) was not the first. Napari et al. (2002), An improved model for ternary nucleation of sulfuric acid–ammonia–water, The Journal of Chemical Physics 116, 4221 was earlier (and there may be others earlier than this).

P2 L32: what does EMAC stand for?

P4 L2: Confusing. Does 214Bi go to 214Po first?

R1-R5: Do all of the species here need to be advected in the model? Many of the

species lifetimes are shorter than typical advection timescales.

P4 L17: What is IGRF? Citation for where these "first 3 coefficients" come from?

P4 L31. The discussion around Eq 2 largely discusses radius, and the variable is "r_um", so I was surprised to see that it was defined as "diameter". Is it actually diameter or is it radius? If it is diameter, it would be better to have the variable be "d" and the use "diameter" for the rest of the discussion.

P5 L3-10: So is growth of small charged particles to larger sizes where they are then "large ions" a loss of small ions then?

P6 L12: "oni"

Table 1 and Section 3.2.1: If I'm correct, this evaluation of the placement NAN before, within, or after GMXe has to do with operator splitting and master timesteps vs. internal GMXe timesteps. By taking NAN out of GMXe nucleation is called on the master timestep and then other aerosol microphysical processes are called in GMXe for the master timestep. When NAN is in GMXe, I'm guessing it can be called more frequently in some internal GMXe timestep. The balance between condensation and nucleation are quite sensitive to the timestep and order of operations, especially when the timestep is similar to or longer than the condensation sink timescale (and this could explain why the results in Figures 5 and 6 are sensitive to the placement of NAN for some cases by not for most). Am I correct about this? If yes, it would make sense to frame the motivation and discussion of Table 1 and Section 3.2.1 around errors due to timesteps and order of operations. Right now, the paper is fairly cryptic as to why the differences arise ("linearization to non-linear processes", but if my hypothesis is correct, I think the explanation is straightforward.

P11 L18: "For two of the stations". It's 3 stations, right?

P12 L10: "Large uncertainties remain, mainly due to the incomplete nature of the implemented nucleation rate parameterizations." This sounds like the authors are saying

that if we just fixed our nucleation rate parameterizations, most of the model uncertainties in aerosol predictions would go away. However, simulating nucleation perfectly would only marginally improve simulations (e.g., Lee, L. A., Pringle, K. J., Reddington, C. L., Mann, G. W., Stier, P., Spracklen, D. V., Pierce, J. R., and Carslaw, K. S.: The magnitude and causes of uncertainty in global model simulations of cloud condensation nuclei, Atmos. Chem. Phys., 13, 8879-8914, doi:10.5194/acp-13-8879-2013, 2013.), Or am I misinterpreting what the authors are trying to say here?

---

## Referee Comment (RC2) · Anonymous Referee #2 · 14 Oct 2018

**Review of the manuscript „Two new submodels for the Modular Earth Submodel System (MESSy): New Aerosol Nucleation (NAN) and small ions (IONS) version 1.0"**

**General comments:**

In the manuscript the authors introduce two new sub-modules into the EMAC/MESSy framework for calculation of new particle formation. NAN calculates nucleation via several pathways and is largely based on experimental results of CLOUD chamber experiments, published previously. Since the new parameterization of nucleation in the NAN depends on atmospheric ions, these were also introduced in MESSy as sub-module IONS.

Although most of the previous CLOUD studies also introduced their new process parameterizations (eg. ion induced ternary nucleation, nucleation involving oxidized organics and pure organic nucleation) into global aerosol models, the coupling with a global chemistry model was not realized yet. Thus, further studies with EMAC/MESSy could also evaluate chemical factors. Moreover, NAN includes several nucleation pathways involving also stabilizing ammonia/ amines and oxidized organics, both neutral and ionic. This approach might be very promising regarding to future usage disentangling dominant pathways as for polluted and pristine environments. In general, the manuscript is structured well and clearly written. Thus, I recommend to accept the manuscript for publication after some minor corrections and clarifications I address in my comments below.

**Specific comments:**

On page 8 you describe the simulations done for testing and evaluating the new sub-modules. Table 2 shows the overview over the model runs, four runs appear there. GMXe, the base run including the new paramterization Dunne et al. (2016) within GMXe, Dunne 1 and Dunne 2 (same parameterization, but calling the sub-module before and after GMXe) and a run named Organic. What is the difference between Dunne 2 and Organic? In the results section, page 10 and 11 the run Organic is not mentioned and not shown in any figure. Please clarify in the text.

In pages 6, 7 and 8 you mention different (or not different?) HOMs. Please clarify the difference between HOM, $HOM_{OH}$, $HOM_{O3}$, $HOMO_{OH}$, $HOMO_{O3}$.

HOMs were not inlcuded in ORACLE and added for this study. How does ORACLE treat these HOMs? Do they also undergo SOA formation driven by ORACLE, outside of nucleation events? How do they interact with pre-existing aerosol?

How much SOA formation results from taking into account the improved nucleation in MESSy? You mentioned the study by Tröstl et al. 2016, where they describe accelerated particle growth due to low and semi volatiles, which are simulated and used in ORACLE.

On page 7 you describe the total nucleation rate and you show particle numbers in the results section. Nevertheless, as you consider various new particle formation pathways, I wounder if you already identified (maybe regionally and temporally) dominant pathways? This would be an interesting point for discussion about competing processes.

**Technical corrections:**

Page 5, line 1: change „The radius of the aerosol particles is provide" to „… is provided".

Page 6, line 10: the first „in" is redundant.

Page 6, line 12: change „oni" to „on".

Page 9 Table 1: In the caption you describe „Position", but in the table the header is „NAN called", please clarify.

Page 19, Table 2: Change „altitude" to „Altitude" for consistency.

Page 25, Figure 6: The caption is wrong according to the run Dunne 2, please change „just before" to „after".

---

## Author Comment (AC1) · 19 Nov 2018

**AR:** We thank Referee 1 for their time and for the useful suggestions and correction. Here we addressed all comments (reporting the original comments). The manuscript as been improved accordingly.

**RC:** P2 L4: Merikanto et al. (2007) was not the first. Napari et al. (2002), An improved model for ternary nucleation of sulfuric acid–ammonia–water, The Journal of Chemical Physics 116, 4221 was earlier (and there may be others earlier than this).

**AR:** We will include the suggested reference, also rephrasing the sentence in the revised version: "Napari et al. (2002) derived a parameterisation of the $H_2SO_4$-$NH_3$-$H_2O$ system based on theoretical calculations and an improved parameterisation was developed by Merikanto et al. (2007)."

**RC:** P2 L32: what does EMAC stand for?

**AR:** EMAC stands for ECHAM/MESSy Atmospheric Chemistry. The definition was given in the abstract but we added again the definition to the mentioned position in the revised manuscript.

**RC:** P4 L2: Confusing. Does 214Bi go to 214Po first?

**AR:** The referee is correct that $^{214}$Bi undergoes a $\beta^-$ decay to $^{214}$Po with a half life time of 20 min. $^{214}$Po $\alpha$ decays almost immediately (half life time 160 $\mu$s). For this reason the aforementioned alpha decay is not explicitly treated by DRADON. The same is true for R5, the decay of $^{210}$Pb to $^{206}$Pb, which goes via $^{210}$Bi and $^{210}$Po to $^{206}$Pb (both $\beta^-$) at life times much shorter than the 22 years for the first step. We also realised that the Proton number of Bi was incorrectly given as 81 instead of the correct 83. We removed proton numbers in the revised manuscript to improve readability. We also indicate now the charge of the $\beta$ decay.

**RC:** R1-R5: Do all of the species here need to be advected in the model? Many of the species lifetimes are shorter than typical advection timescales.

**AR:** The reactions R1-R5 are part of the DRADON submodel, an already existing submodel in MESSy. Advection for each singular tracer can be (de)activated via namelist, and therefore user-dependent. Indeed, for such tracers, advection could be avoided as already done for many other species (e.g. $O_3^P$).

[Figure]

**RC:** P4 L17: What is IGRF? Citation for where these "first 3 coefficients" come from?

**AR:** IGRF is the International Geomagnetic Reference Field. We added this missing definition to the text and added the relevant citation.

**RC:** P4 L31. The discussion around Eq 2 largely discusses radius, and the variable is "r_um", so I was surprised to see that it was defined as "diameter". Is it actually diameter or is it radius? If it is diameter, it would be better to have the variable be "d" and the use "diameter" for the rest of the discussion.

**AR:** Eq 2 is parameterised for a radius in micro meter. For clarity, we use now $d_{\mu m}/2$ in Eq 2 and 3 and changed all text to diameter. We hope this avoids confusion.

**RC:** P5 L3-10: So is growth of small charged particles to larger sizes where they are then "large ions" a loss of small ions then?

**AR:** It is a loss of ions which can nucleate. An ion that nucleated is an intermediate size ion. For ions with a diameter of up to 10 nm the ion-ion recombination coefficient is still around 1.6e-6 $cm^3$ $s^{-1}$. Assuming absence of aerosol particles, or any other loss, and an ion pair production rate of 5 i.p. $cm^{-3}$ $s^{-1}$ at ground level (Figure 4 a) the steady state ion pair concentration would be around 1800 i.p. $cm^{-3}$. This would mean an ion life time of 350 s, roughly 6 minutes. Losses to aerosol particles will change this picture slightly but will also lower the overall lifetime of small ions. All factors included the lifetime of small ion clusters is very short while the time for an ion cluster to grow to a diameter larger than 10 nm is in most circumstances longer, a typical rule of thumb is 1 nm/h for a $[H_2SO_4]$ = 1e7 $cm^{-3}$. The size of ions becomes important when the conductivity of air is calculated. For such a calculation a more detailed ion aerosol model is required that describes ions size resolved.

[Figure]

**RC:** P6 L12: "oni"

**AR:** We corrected it to "on".

**RC:** Table 1 and Section 3.2.1: If I'm correct, this evaluation of the placement NAN be- fore, within, or after GMXe has to do with operator splitting and master timesteps vs. internal GMXe timesteps. By taking NAN out of GMXe nucleation is called on the master timestep and then other aerosol microphysical processes are called in GMXe for the master timestep. When NAN is in GMXe, I'm guessing it can be called more frequently in some internal GMXe timestep. The balance between condensation and nucleation are quite sensitive to the timestep and order of operations, especially when the timestep is similar to or longer than the condensation sink timescale (and this could explain why the results in Figures 5 and 6 are sensitive to the placement of NAN for some cases by not for most). Am I correct about this? If yes, it would make sense to frame the motivation and discussion of Table 1 and Section 3.2.1 around errors due to timesteps and order of operations. Right now, the paper is fairly cryptic as to why the differences arise ("linearization to non-linear processes", but if my hypothesis is correct, I think the explanation is straightforward.

**AR:** The referee is partially right. We added the following paragraph to the revised MSs, to motivate the analysis and Section 3.2.1: "Nucleation rates typical follow a power law with respect to vapour concentrations, see for example Kashchiev (1982) and Oxtoby and Kashchiev (1994). Therefore small changes in the vapour concentration, here $H_2SO_4$ and $NH_3$, can have a large influence on the nucleation rate. Condensation proceeds typically faster than nucleation, it is reasonable to place the nucleation after the condensation in a time step. Therefore, the original implementation of GMXe calculates nucleation after it calculates the amount of vapour that condensed on aerosol particles. There is no internal shorter time step in GMXe. However, condensation is not the only process affecting vapour concentrations, or particle concentration. Therefore aerosol particle concentrations are also sensitive to the placement of GMXe within MESSy's

interface layer. Unfortunately, making microphysical processes available for as many submodels and potential users as possible is best achieved as a submodel, as MESSy has currently no unified interface definition for sub-submodels, i.e. a submodel of a submodel. Therefore, implementation of NAN and IONS as submodel was preferred as both models can be called independently of the choice of other submodels."

**RC:** P11 L18: "For two of the stations". It's 3 stations, right?

**AR:** The referee is correct, we show 3 stations. We corrected it and also added now Mace Head into the text.

**RC:** P12 L10: "Large uncertainties remain, mainly due to the incomplete nature of the im- plemented nucleation rate parameterizations." This sounds like the authors are saying that if we just fixed our nucleation rate parameterizations, most of the model uncer- tainties in aerosol predictions would go away. However, simulating nucleation perfectly would only marginally improve simulations (e.g., Lee, L. A., Pringle, K. J., Redding- ton, C. L., Mann, G. W., Stier, P., Spracklen, D. V., Pierce, J. R., and Carslaw, K. S.: The magnitude and causes of uncertainty in global model simulations of cloud conden- sation nuclei, Atmos. Chem. Phys., 13, 8879-8914, doi:10.5194/acp-13-8879-2013, 2013.), Or am I misinterpreting what the authors are trying to say here?

**AR:** The uncertainties of the nucleation rates are due to the incomplete nature of the parameterisation. As the referee mentioned in their reply: ".. for the overall aerosol pre- dictions many factors contribute to the overall uncertainty." We rephrase the sentence to avoid misinterpretation.

[Figure]

**References**

Kashchiev, D.: On the relation between nucleation work, nucleus size, and nucleation rate, The Journal of Chemical Physics, 76, 5098–5102, doi:10.1063/1.442808, 1982.

Merikanto, J., Napari, I., Vehkamäki, H., Anttila, T., and Kulmala, M.: New parameterization of sulfuric acid-ammonia-water ternary nucleation rates at tropospheric conditions, Journal of Geophysical Research: Atmospheres, 112, n/a–n/a, doi:10.1029/2006JD007977, http://dx.doi. org/10.1029/2006JD007977, d15207, 2007.

Napari, I., Noppel, M., Vehkamäki, H., and Kulmala, M.: Parametrization of ternary nucleation rates for H2SO4-NH3-H2O vapors, Journal of Geophysical Research: Atmospheres, 107, AAC 6–1–AAC 6–6, doi:10.1029/2002JD002132, http://dx.doi.org/10.1029/2002JD002132, 4381, 2002.

Oxtoby, D. W. and Kashchiev, D.: A general relation between the nucleation work and the size of the nucleus in multicomponent nucleation, The Journal of Chemical Physics, 100, 7665–7671, doi:10.1063/1.466859, 1994.

---

## Author Comment (AC2) · 19 Nov 2018

**AR:** We thank Referee 2 for their time and for the useful suggestions and correction. Here we addressed all comments (reporting the original comments). The manuscript has been improved accordingly.

**General comments:**

**RC:** In the manuscript the authors introduce two new sub-modules into the EMAC/MESSy framework for calculation of new particle formation. NAN calculates nucleation via several pathways and is largely based on experimental results of CLOUD chamber experiments, published previously. Since the new parameterization of nucleation in the NAN depends on atmospheric ions, these were also introduced in MESSy as sub-module IONS. Although most of the previous CLOUD studies also introduced their new process parameterizations (eg. ion induced ternary nucleation, nucleation involving oxidized organics and pure organic nucleation) into global aerosol models, the coupling with a global chemistry model was not realized yet. Thus, further studies with EMAC/MESSy could also evaluate chemical factors. Moreover, NAN includes several nucleation pathways involving also stabilizing ammonia/ amines and oxidized organics, both neutral and ionic. This approach might be very promising regarding to future usage disentangling dominant pathways as for polluted and pristine environments. In general, the manuscript is structured well and clearly written. Thus, I recommend to accept the manuscript for publication after some minor corrections and clarifications I address in my comments below.

**AR**: We thank the referee for the positive comments on the manuscript. Further studies utilising the new submodels and improved reactions will certainly be conducted and looked at these topics in more detail.

**Specific comments:**

**RC:** On page 8 you describe the simulations done for testing and evaluating the new sub-modules. Table 2 shows the overview over the model runs, four runs appear there. GMXe, the base run including the new paramterization Dunne et al. (2016) within

[Figure]

GMXe, Dunne 1 and Dunne 2 (same parameterization, but calling the sub-module before and after GMXe) and a run named Organic. What is the difference between Dunne 2 and Organic? In the results section, page 10 and 11 the run Organic is not mentioned and not shown in any figure. Please clarify in the text.

**AR:** We assume the referee meant Table 1, based on the text of their comment. The run labelled organic is shown in Figure 7 and 8. The organic nucleation was not implemented in GMXe and the effects of calling nucleation outside of GMXe was tested only with the inorganic nucleation, as $H_2SO_4$ and $NH_3$ are already part of GMXe. We removed the organic entry from Table 1 to avoid this confusion.

**RC:** In pages 6, 7 and 8 you mention different (or not different?) HOMs. Please clarify the difference between HOM, HOMOH, HOMO3, HOMOOH, HOMOO3.

**AR:** $HOMO_{OH}$ and $HOMO_{O_3}$ are typos. We corrected them in the revised version of the manuscript. $HOM_{OH}$ are products of monoterpene oxidation by OH radicals that can nucleate. $HOM_{O_3}$ are products of monoterpene oxidation by $O_3$. HOM without any subscript is the sum of $HOM_{OH}$ and $HOM_{O_3}$ (page 7, line 2 of the original discussion paper).

**RC:** HOMs were not inlcuded in ORACLE and added for this study. How does ORACLE treat these HOMs?

**AR:** Species that form SOA can be added to ORACLE via namelists. The chemistry of $HOM_{O_3}$ and $HOM_{OH}$ formation is described in R6 and R7, this reaction was added to the ORACLE chemistry as described in Tsimpidi et al. (2014). These HOMs are then added to an ORACLE volatility bin and treated in the same way as other species in ORACLE according to their vapour pressure (page 8 line2 31-32).

**RC:** Do they also undergo SOA formation driven by ORACLE, outside of nucleation

events? How do they interact with pre-existing aerosol?

**AR:** If aerosol particles are present, HOMs will be partitioned between gas phase and particle phase by ORACLE.

**RC:** How much SOA formation results from taking into account the improved nucleation in MESSy? You mentioned the study by Tröstl et al. 2016, where they describe accelerated particle growth due to low and semi volatiles, which are simulated and used in ORACLE. On page 7 you describe the total nucleation rate and you show particle numbers in the results section. Nevertheless, as you consider various new particle formation pathways, I wounder if you already identified (maybe regionally and temporally) dominant pathways? This would be an interesting point for discussion about competing processes.

**AR:**The aforementioned questions will be subject of more detailed studies with these new submodels.

**Technical corrections:**

**RC:** Page 5, line 1: change "The radius of the aerosol particles is provide" to "... is provided".

**AR:** We corrected this.

**RC:** Page 6, line 10: the first "in" is redundant.

**AR:** We corrected the sentence.

**RC:** Page 6, line 12: change "oni" to "on".

**AR:** We fixed this mistake.

**RC:** Page 9 Table 1: In the caption you describe "Position", but in the table the header is "NAN called", please clarify.

**AR:** We corrected the header. It is now in accordance with the text "Position".

**RC:** Page 19, Table 2: Change "altitude" to "Altitude" for consistency.

**AR:** We changed the text and also added the information that altitude is in m.

**RC:** Page 25, Figure 6: The caption is wrong according to the run Dunne 2, please change "just before" to "after".

**AC**: We corrected the sentence.

**References**

Tsimpidi, A. P., Karydis, V. A., Pozzer, A., Pandis, S. N., and Lelieveld, J.: ORACLE (v1.0): module to simulate the organic aerosol composition and evolution in the atmosphere, Geoscientific Model Development, 7, 3153–3172, doi:10.5194/gmd-7-3153-2014, http://www.geosci-model-dev.net/7/3153/2014/, 2014.

---

## Author Response (AR1)

**Combined Response**

Sebastian Ehrhart et al

We present here the responses to the Referee comments and a version of the manuscript with changes indicated in one file. Referee comments start with **RC**, Author response with **AR** and where changes were made to the manuscript a pargraph starting with **Change** is given. The responses are grouped as list items for easier navigation. The revised manuscript starts on page 10.

**Response to Referee 1**

- **RC:** P2 L4: Merikanto et al. (2007) was not the first. Napari et al. (2002), An improved model for ternary nucleation of sulfuric acid–ammonia–water, The Journal of Chemical Physics 116, 4221 was earlier (and there may be others earlier than this).

  **AR:** We will include the suggested reference, also rephrasing the sentence in the revised version: "Napari et al. (2002) derived a parameterisation of the $H_2SO_4$-$NH_3$-$H_2O$ system based on theoretical calculations and an improved parameterisation was developed by Merikanto et al. (2007)."

  **Change:**  Napari et al. (2002) derived a parameterisation of the $H_2SO_4$-$NH_3$-$H_2O$ system based on theoretical calculations and an improved parameterisation was developed by Merikanto et al. (2007).

- **RC:** P2 L32: what does EMAC stand for?

  **AR:** EMAC stands for ECHAM/MESSy Atmospheric Chemistry. The definition was given in the abstract but we added again the definition to the mentioned position in the revised manuscript.

  **Change:** In this work the implementation of the CLOUD based parameterisations into the Modular Earth Submodel System (MESSy) is described, as well as their application in the  ECHAM/MESSy Atmospheric Chemistry (EMAC) GCM.

- **RC:** P4 L2: Confusing. Does 214Bi go to 214Po first?

  **AR:** The referee is correct that $^{214}$Bi undergoes a $\beta^-$ decay to $^{214}$Po with a half life time of 20 min. $^{214}$Po $\alpha$ decays almost immediately (half life time 160 $\mu$s). For this reason the aforementioned alpha decay is not explicitly treated by DRADON. The same is true for R5, the decay of $^{210}$Pb to $^{206}$Pb, which goes via $^{210}$Bi and $^{210}$Po to $^{206}$Pb (both $\beta^-$) at

life times much shorter than the 22 years for the first step. We also realised that the Proton number of Bi was incorrectly given as 81 instead of the correct 83. We removed proton numbers in the revised manuscript to improve readability. We also indicate now the charge of the $\beta$ decay.

**Change:**

$$^{222}_{86}\mathrm{Rn} \xrightarrow{3.8\mathrm{d}} {}^{218}_{84}\mathrm{Po} + \alpha\ 5.59\mathrm{MeV} \tag{R1}$$

$$^{218}_{84}\mathrm{Po} \xrightarrow{180\mathrm{s}} {}^{214}_{82}\mathrm{Pb} + \alpha\ 6.12\mathrm{MeV} \tag{R2}$$

$$^{214}_{82}\mathrm{Pb} \xrightarrow{27\mathrm{min}} {}^{214}_{81}\mathrm{Bi} + \beta^- \ 1.02\mathrm{MeV} \tag{R3}$$

$$^{214}_{81}\mathrm{Bi} \xrightarrow{20\mathrm{min}} \mathrm{Bi} \xrightarrow[\text{via } ^{214}\mathrm{Po}]{20\mathrm{min}} {}^{210}_{82}\mathrm{Pb} + \beta^- + \alpha\ (7.88 + 3.27)\mathrm{MeV} \tag{R4}$$

$$^{210}_{82}\mathrm{Pb} \rightarrow ... \xrightarrow{22.3\mathrm{y}} {}^{206}_{82}\mathrm{Pb} + \alpha \tag{R5}$$

- **RC:** R1-R5: Do all of the species here need to be advected in the model? Many of the species lifetimes are shorter than typical advection timescales.

  **AR:** The reactions R1-R5 are part of the DRADON submodel, an already existing submodel in MESSy. Advection for each singular tracer can be (de)activated via namelist, and therefore user-dependent. Indeed, for such tracers, advection could be avoided as already done for many other species (e.g. $O_3^P$).

- **RC:** P4 L17: What is IGRF? Citation for where these "first 3 coefficients" come from?

  **AR:** IGRF is the International Geomagnetic Reference Field. We added this missing definition to the text and added the relevant citation.

  **Change:** The geomagnetic cut off rigidity uses the first 3 coefficients of the  International Geomagnetic Reference Field (IGRF) (Thébault et al., 2015) coefficients of Earth's magnetic field.

- **RC:** P4 L31. The discussion around Eq 2 largely discusses radius, and the variable is "r_um", so I was surprised to see that it was defined as "diameter". Is it actually diam- eter or is it radius? If it is diameter, it would be better to have the variable be "d" and the use "diameter" for the rest of the discussion.

  **AR:** Eq 2 is parameterised for a radius in micro meter. For clarity, we use now $d_{\mu m}/2$ in Eq 2 and 3 and changed all text to diameter. We hope this avoids confusion.

**Change:** For particles with a diameter larger than 20 nm, the expression

$$k_a = 4.36 \cdot 10^{-5}  \frac{d_{\mu m}}{2} - 9.2 \cdot 10^{-8} \tag{1}$$

from Hoppel (1985) is used to calculate the attachment rate coefficient.  $d_{\mu m}$ is the aerosol particle diameter in $\mu m$. For particles smaller than this  Tinsley and Zhou (2006) provided,

$$\log_{10} k_a = 1.243 \log_{10}  \frac{d_{\mu m}}{2} - 3.978 \tag{2}$$

as extrapolation for nucleation mode particles. The  size of the aerosol particles is  provided by aerosol submodels such as GMXe.

- **RC:** P5 L3-10: So is growth of small charged particles to larger sizes where they are then "large ions" a loss of small ions then?

  **AR:** It is a loss of ions which can nucleate. An ion that nucleated is an intermediate size ion. For ions with a diameter of up to 10 nm the ion-ion recombination coefficient is still around 1.6e-6 $cm^3 \ s^{-1}$. Assuming absence of aerosol particles, or any other loss, and an ion pair production rate of 5 i.p. $cm^{-3} \ s^{-1}$ at ground level (Figure 4 a) the steady state ion pair concentration would be around 1800 i.p. $cm^{-3}$. This would mean an ion life time of 350 s, roughly 6 minutes. Losses to aerosol particles will change this picture slightly but will also lower the overall lifetime of small ions. All factors included the lifetime of small ion clusters is very short while the time for an ion cluster to grow to a diameter larger than 10 nm is in most circumstances longer, a typical rule of thumb is 1 nm/h for a $[H_2SO_4] = $ 1e7 $cm^{-3}$. The size of ions becomes important when the conductivity of air is calculated. For such a calculation a more detailed ion aerosol model is required that describes ions size resolved.

- **RC:** P6 L12: "oni"

  **AR:** We corrected it to "on".

  **Change:** Dunne et al. (2016) give a scaling factor dependent on the relative humidity as fraction, $RH$, and temperature, $T$, in Kelvin

  $$f_{RH} = 1 + c_1 \left( RH - 0.38 \right) + c_2 \left( RH - 0.38 \right)^3 \left( T - 208K \right)^2 , \tag{3}$$

  with $c_1 = 1.5$ and $c_2 = 0.045 \ K^{-2}$. However, this scaling factor is more of an ad hoc solution and based  on very few measurements. The overall effect of this scaling is described as relatively small in Dunne et al. (2016) and is not used here.

- **RC:** Table 1 and Section 3.2.1: If I'm correct, this evaluation of the placement NAN be- fore, within, or after GMXe has to do with operator splitting and master timesteps vs. internal GMXe timesteps. By taking NAN out of GMXe nucleation is called on the master timestep and then other aerosol microphysical processes are called in GMXe for the

master timestep. When NAN is in GMXe, I'm guessing it can be called more frequently in some internal GMXe timestep. The balance between condensation and nucleation are quite sensitive to the timestep and order of operations, especially when the timestep is similar to or longer than the condensation sink timescale (and this could explain why the results in Figures 5 and 6 are sensitive to the placement of NAN for some cases by not for most). Am I correct about this? If yes, it would make sense to frame the motivation and discussion of Table 1 and Section 3.2.1 around errors due to timesteps and order of operations. Right now, the paper is fairly cryptic as to why the differences arise ("linearization to non-linear processes"), but if my hypothesis is correct, I think the explanation is straightforward.

**AR:** The referee is partially right. We added the following paragraph to the revised MSs, to motivate the analysis and Section 3.2.1: "Nucleation rates typical follow a power law with respect to vapour concentrations, see for example Kashchiev (1982) and Oxtoby and Kashchiev (1994). Therefore small changes in the vapour concentration, here $H_2SO_4$ and $NH_3$, can have a large influence on the nucleation rate. Condensation proceeds typically faster than nucleation, it is reasonable to place the nucleation after the condensation in a time step. Therefore, the original implementation of GMXe calculates nucleation after it calculates the amount of vapour that condensed on aerosol particles. There is no internal shorter time step in GMXe. However, condensation is not the only process affecting vapour concentrations, or particle concentration. Therefore aerosol particle concentrations are also sensitive to the placement of GMXe within MESSy's interface layer. Unfortunately, making microphysical processes available for as many submodels and potential users as possible is best achieved as a submodel, as MESSy has currently no unified interface definition for sub-submodels, i.e. a submodel of a submodel. Therefore, implementation of NAN and IONS as submodel was preferred as both models can be called independently of the choice of other submodels."

**Change:** Nucleation rates typical follow a power law with respect to vapour concentrations, see for example Kashchiev (1982) and Oxtoby and Kashchiev (1994). Therefore small changes in the vapour concentration, here $H_2SO_4$ and $NH_3$, can have a large influence on the nucleation rate. Condensation proceeds typically faster than nucleation, it is reasonable to place the nucleation after the condensation in a time step. Therefore, the original implementation of GMXe calculates nucleation after it calculates the amount of vapour that condensed on aerosol particles. There is no internal shorter time step in GMXe. However, condensation is not the only process affecting vapour concentrations, or particle concentration. Therefore aerosol particle concentrations are also sensitive to the placement of GMXe within MESSy's interface layer. Unfortunately, making microphysical processes available for as many submodels and potential users as possible is best achieved as a submodel, as MESSy has currently no unified interface definition for sub-submodels, i.e. a submodel of a submodel. Therefore, implementation of NAN and IONS as submodel was preferred as both models can be called independently of the choice of other submodels.

– **RC:** P11 L18: "For two of the stations". It's 3 stations, right?

**AR:** The referee is correct, we show 3 stations. We corrected it and also added now Mace Head into the text.

**Change:** For  three of the stations, the monthly distributions of particle concentrations are shown ... Nevertheless the model catches certain seasonality for some stations, shown here for Hyytiälä and to a lesser degree Mace Head

- **RC:** P12 L10: "Large uncertainties remain, mainly due to the incomplete nature of the im- plemented nucleation rate pa- rameterizations." This sounds like the authors are saying that if we just fixed our nucleation rate parameterizations, most of the model uncer- tainties in aerosol predictions would go away. However, simulating nucleation perfectly would only marginally improve simulations (e.g., Lee, L. A., Pringle, K. J., Redding- ton, C. L., Mann, G. W., Stier, P., Spracklen, D. V., Pierce, J. R., and Carslaw, K. S.: The magnitude and causes of uncertainty in global model simulations of cloud conden- sation nuclei, Atmos. Chem. Phys., 13, 8879-8914, doi:10.5194/acp-13-8879-2013, 2013.), Or am I misinter- preting what the authors are trying to say here?

  **AR:** The uncertainties of the nucleation rates are due to the incomplete nature of the parameterisation. As the referee mentioned in their reply: ".. for the overall aerosol predictions many factors contribute to the overall uncertainty." We rephrase the sentence to avoid misinterpretation.

**Responses to Referee 2**

- **RC:** In the manuscript the authors introduce two new sub-modules into the EMAC/MESSy framework for calculation of new particle formation. NAN calculates nucleation via several pathways and is largely based on experimental results of CLOUD chamber experiments, published previously. Since the new parameterization of nucleation in the NAN depends on atmospheric ions, these were also introduced in MESSy as sub-module IONS. Although most of the previous CLOUD studies also introduced their new process parameterizations (eg. ion induced ternary nucleation, nucleation involving oxidized organics and pure organic nucleation) into global aerosol models, the coupling with a global chemistry model was not realized yet. Thus, further studies with EMAC/MESSy could also evaluate chemical factors. Moreover, NAN includes several nucleation pathways involving also stabilizing ammonia/ amines and oxidized organics, both neutral and ionic. This approach might be very promising regarding to future usage disentangling dominant pathways as for polluted and pristine environments. In general, the manuscript is structured well and clearly written. Thus, I recommend to accept the manuscript for publication after some minor corrections and clarifications I address in my comments below.

  **AR**: We thank the referee for the positive comments on the manuscript. Further studies utilising the new submodels and improved reactions will certainly be conducted and looked at these topics in more detail.

- **RC:** On page 8 you describe the simulations done for testing and evaluating the new sub-modules. Table 2 shows the overview over the model runs, four runs appear there. GMXe, the base run including the new paramterization Dunne et al. (2016) within GMXe, Dunne 1 and Dunne 2 (same parameterization, but calling the sub-module before and after GMXe) and a run named Organic. What is the difference between Dunne 2 and Organic? In the results section, page 10 and 11 the run Organic is not mentioned and not shown in any figure. Please clarify in the text.

  **AR:** We assume the referee meant Table 1, based on the text of their comment. The run labelled organic is shown in Figure 7 and 8. The organic nucleation was not implemented in GMXe and the effects of calling nucleation outside of

GMXe was tested only with the inorganic nucleation, as $H_2SO_4$ and $NH_3$ are already part of GMXe. We removed the organic entry from Table 1 to avoid this confusion.

**Change:** *See table 1 in the attached document.*

– **RC:** In pages 6, 7 and 8 you mention different (or not different?) HOMs. Please clarify the difference between HOM, HOMOH, HOMO3, HOMOOH, HOMOO3.

**AR:** $HOMO_{OH}$ and $HOMO_{O_3}$ are typos. We corrected them in the revised version of the manuscript. $HOM_{OH}$ are products of monoterpene oxidation by OH radicals that can nucleate. $HOM_{O_3}$ are products of monoterpene oxidation by $O_3$. HOM without any subscript is the sum of $HOM_{OH}$ and $HOM_{O_3}$ (page 7, line 2 of the original discussion paper).

**Change:** As mentioned in section 2.3, the terpene oxidation product is split into the product of ozonolysis of terpenes and oxidation of terpenes with OH radicals, leading to

$$LTERP + O_3 \rightarrow HOM_{O_3} \tag{R6}$$

and

$$LTERP + OH \rightarrow HOM_{OH} \tag{R7}$$

as the reactions of the aerosol precursor gas. The lumped terpene tracer, LTERP, is based on terpene emissions from Tsimpidi et al. (2014). The gas to particle phase partitioning of the added organic species is calculated by ORACLE (Tsimpidi et al., 2014). A saturation vapour pressure of $2 \cdot 10^{-2} \, \mu gm^{-3}$ was assumed for $\underline{HOM_{OH}}$ and $\underline{HOM_{O_3}}$. This places the saturation vapour pressure within the LVOC regime as described in Tröstl et al. (2016).

– **RC:** HOMs were not inlcuded in ORACLE and added for this study. How does ORACLE treat these HOMs?

**AR:** Species that form SOA can be added to ORACLE via namelists. The chemistry of $HOM_{O_3}$ and $HOM_{OH}$ formation is described in R6 and R7, this reaction was added to the ORACLE chemistry as described in Tsimpidi et al. (2014). These HOMs are then added to an ORACLE volatility bin and treated in the same way as other species in ORACLE according to their vapour pressure (page 8 line2 31-32).

– **RC:** Do they also undergo SOA formation driven by ORACLE, outside of nucleation events? How do they interact with pre-existing aerosol?

**AR:** If aerosol particles are present, HOMs will be partitioned between gas phase and particle phase by ORACLE.

– **RC:** How much SOA formation results from taking into account the improved nucleation in MESSy? You mentioned the study by Tröstl et al. 2016, where they describe accelerated particle growth due to low and semi volatiles, which are simulated and used in ORACLE. On page 7 you describe the total nucleation rate and you show particle numbers in the results section. Nevertheless, as you consider various new particle formation pathways, I wounder if you already

identified (maybe regionally and temporally) dominant pathways? This would be an interesting point for discussion about competing processes.

**AR:**The aforementioned questions will be subject of more detailed studies with these new submodels.

**Technical corrections:**

- **RC:** Page 5, line 1: change „The radius of the aerosol particles is provide" to „... is provided".

  **AR:** We corrected this.

  **Change:** The  size of the aerosol particles is  provided by aerosol submodels such as GMXe.

- **RC:** Page 6, line 10: the first „in" is redundant.

  **AR:** We corrected the sentence.

  **Change:** Dunne et al. (2016) give a scaling factor dependent on the relative humidity as fraction, $RH$, and temperature, $T$, in Kelvin ..

- **RC:** Page 6, line 12: change „oni" to „on".

  **AR:** We fixed this mistake.

  **Change:** However, this scaling factor is more of an ad hoc solution and based  on very few measurements. The overall effect of this scaling is described as relatively small in Dunne et al. (2016) and is not used here.

- **RC:** Page 9 Table 1: In the caption you describe „Position", but in the table the header is „NAN called", please clarify.

  **AR:** We corrected the header. It is now in accordance with the text "Position".

  **Change:** *See table 1 in attached document.*

- **RC:** Page 19, Table 2: Change „altitude" to „Altitude" for consistency.

  **AR:** We changed the text and also added the information that altitude is in m.

  **Change:** Station coordinates taken from the EBAS data files. *Altitude* is given in m. *Station names in Italic indicate locations with ion measurements.*

- **RC:** Page 25, Figure 6: The caption is wrong according to the run Dunne 2, please change „just before" to „after".

  **AC:** We corrected the sentence.

[revised manuscript text omitted]